# Dentin Particulate for Bone Regeneration: An In Vitro Study

**DOI:** 10.3390/ijms23169283

**Published:** 2022-08-18

**Authors:** Giulia Brunello, Federica Zanotti, Gerard Scortecci, Lari Sapoznikov, Stefano Sivolella, Barbara Zavan

**Affiliations:** 1Department of Neurosciences, Dentistry Section, University of Padova, 35122 Padova, Italy; 2Department of Oral Surgery, University Clinic Düsseldorf, 40225 Düsseldorf, Germany; 3Department of Translational Medicine, University of Ferrara, 44121 Ferrara, Italy; 4Institut Universitaire de la Face et du Cou—de Nice et Centre Lacassagne (IUFC), 06100 Nice, France; 5Independent Researcher, Tel Aviv 6997801, Israel

**Keywords:** odontogenesis, dentin, dentinogenesis, embryonic, ivory

## Abstract

The aim of this in vitro study was to investigate the commitment and behavior of dental pulp stem cells (DPSCs) seeded onto two different grafting materials, human dentin particulate (DP) and deproteinized bovine bone matrix (BG), with those cultured in the absence of supplements. Gene expression analyses along with epigenetic and morphological tests were carried out to examine odontogenic and osteogenic differentiation and cell proliferation. Compressive testing of the grafting materials seeded with DPSCs was performed as well. DPSC differentiation into odontoblast-like cells was identified from the upregulation of odontogenic markers (DSPP and MSX) and osteogenic markers (RUNX2, alkaline phosphatase, osteonectin, osteocalcin, collagen type I, bmp2, smad5/8). Epigenetic tests confirmed the presence of miRNAs involved in odontogenic or osteogenic commitment of DPSCs cultured for up to 21 days on DP. Compressive strength values obtained from extracellular matrix (ECM) synthesized by DPSCs showed a trend of being higher when seeded onto DP than onto BG. High expression of VEGF factor, which is related to angiogenesis, and of dentin sialoprotein was observed only in the presence of DP. Morphological analyses confirmed the typical phenotype of adult odontoblasts. In conclusion, the odontogenic and osteogenic commitment of DPSCs and their respective functions can be achieved on DP, which enables exceptional dentin and bone regeneration.

## 1. Introduction

Alveolar bone deficiencies due to tooth loss, trauma, or disease are treated using various techniques, such as ridge augmentation or sinus floor elevation, with the ultimate aim of dental implant-supported rehabilitation. The choice of the most appropriate bone substitute from among a variety of autologous, homologous, heterologous, and synthetic bone grafting materials is challenging [1,2,3,4,5]. When tooth extraction is necessary, a tooth-derived bone graft may be an option, given its autologous origin and related favorable clinical and histological outcomes. The resemblance between the organic and inorganic components of the bone and tooth and evidence of the osteoinductivity and osteoconductivity of autologous tooth particles confirm the appropriateness of this graft material [6,7,8,9,10,11,12].

Many in vivo animal and human studies have been published on the use of autologous dentin particulate (DP), showing its efficacy in terms of minimal graft resorption, limited postoperative complications, ease of placement, and stability on follow-up of dental implants [13,14,15,16,17,18,19,20,21]. Histology has shown that new bone is deposited directly on the surface of the DP graft and a mineralized matrix connection forms between the host bone, the newly formed bone, and the graft material (ankylosis) [15,16,18,19]. During healing, the relative proportion of bone increases while the proportion of DP decreases over time [13,15].

It has been demonstrated that adult mesenchymal stem cells, in particular dental pulp stem cells (DPSCs), cultured on a dentin-derived bio-instructive scaffold can differentiate and secrete an extracellular matrix (ECM) capable of driving osteogenesis and dentinogenesis both in vitro and in vivo [22]. The structural and biochemical composition of DP is similar to that of natural bone. It might promote a mechanical induction that positively affects cell differentiation and proliferation as well as the organization of the ECM. Decellularized ECM materials have proven to be ideal candidates for tissue engineering applications [23].

In light of the above findings, autologous dentin from teeth extracted from the same patient could serve as a good support for bone regeneration. The reasons for its use as a bone substitute include the graft’s acceptance by the host; its very slow resorption (up to 10 years); a rapid and strong dentin- host bone fusion (within weeks), enabling subsequent treatments to proceed; mechanical properties fully matching those of intramembranous cortical jaw bone [24,25]; and the formation of a bone–dentin conglomerate. Beginning in the 1980s, dentin has been tested as a biomaterial for inducing bone formation in several clinical studies, with promising results [26,27,28].

Bone substitutes such as deproteinized bovine bone matrices are extensively used in clinical practice; however, the presence of collagen and non-collagenous proteins (NCPs), including growth factors, in the DP may lead to higher bioactivity [29,30,31,32,33,34].

Autologous dentin grafts can be obtained at the dental clinic immediately after a tooth extraction (chairside method) [30,31]. Alternatively, extracted teeth can be sent to tissue banks, where they are turned into demineralized dentin matrix grafts and then returned to the dentist for use in clinical applications [32,33]. This latter procedure is time-consuming, and the grafting material is not immediately ready for use, whereas with the chairside method the tooth-derived graft can be prepared without delay and used in the same surgical session.

In light of all these considerations, the aim of the present study was to investigate how DP-derived mechanical induction might affect DPSC odontogenic and osteogenic properties when compared to deproteinized bovine bone matrices. A secondary aim was to assess the in vitro ECM mechanical properties of dental particulate.

## 2. Results

### 2.1. Characterization of DPSCs by Flow Cytometry

Cell surface antigen characterization was performed on DPSCs by flow cytometry. DPSCs were found to be positive for the MSC markers CD29, CD73, CD90, and CD105, and negative for the hematopoietic markers CD14 and CD45 (Figure 1).

### 2.2. Cell Proliferation

The ability to proliferate on DP was examined by means of DPSC seeding. Proliferation on DP was assessed using the O.D. value derived from seeding on BG, and no significant differences emerged between the two types of granules at each time point (Figure 2), confirming that both BG and DP supported cell proliferation equally well.

### 2.3. Gene Expression of Odontoblastic and Osteoblastic Markers

The grafting materials’ ability to support odontogenic and/or osteogenic commitment was ascertained by measuring gene expression after 21 days of culturing (Figure 3). Dentin sialoprotein precursor (DSPP), a gene related to the odontogenic lineage, was overexpressed when DPSCs were cultured on DP, while they were not detected when cells were seeded on BG (*p* < 0.001). A similar expression was found for the muscle segment homeobox gene family (MSX), which is involved in the calcification process. By contrast, genes related to osteogenic commitment, such as runt-related transcription factor 2 (RUNX2), osteocalcin (OC), osteonectin (ON), alkaline phosphatase (ALP), and collagen type I (COL1A1), were expressed on both grafting materials.

### 2.4. Epigenetics

The influence of the extracellular environment on the final destination of DPSCs was examined by means of the gene expression of epigenetic markers related to odontogenesis and osteogenesis. Two groups of genes involved in specific pathways capable of affecting the final commitment of stem cells, namely, odontoblastic or osteoblastic, were selected for our samples. The genes involved in the odontoblastic phenotype were CREB and SMAD1. The genes selected for proteins capable of inducing an osteoblastic phenotype were WNT, RAC, CTNNB1, TGF, SMAD4, SMAD5, SMAD 8, BMP2, and p38MAPK. As shown in Figure 4A, DPSCs were only able to express genes related to an odontoblastic phenotype after culturing on DP, and they showed a strong SMAD1 expression. If cells were seeded on BG, significantly lower expression of the same gene was observed. Regarding the osteogenic lineage, the cells were able to express genes related to an osteogenic-like phenotype on DP to a larger extent than on BG.

The second group of genes considered were those related to epigenetics (Figure 4B,C), such as miRNA 26/27/32/586/885-5/smad5 and smad8 for the osteoblastic lineage and miRNA 218 and miRNA 200 for the odontogenic lineage. In this case, it was clear that DP supported the expression of miRNAs involved in odontogenesis and osteogenesis, whereas BG only supported an osteoblastic commitment, and to a lower extent. The increase of bmp2 and smad5/8 provide further evidence that the 3D scaffolds support osteogenic commitment in any case.

### 2.5. Extracellular Matrix and Adhesion Proteins

Gene expression related to extracellular matrix (ECM) and adhesion proteins was performed (Figure 5a). DPSCs cultured up to 21 days on DP or BG showed similar expression related to structural proteins such as collagen type I, vitronectin, and serpin. On the other hand, higher expression values were found in the DP group for intracellular proteins related to extracellular environment, such as actin, and cell adhesion proteins such as integrins, anti-inflammatory interleukin-10 (IL-10), and vascular endothelial growth factor-A (VEGF), which is a fundamental angiogenetic growth factor, as well as TGF and fibroblast growth factors (FGF).

Soluble proteins such as collagene type I and vitronectin, which are related to the extracellular matrix (ECM), enzymes related to ECM remodelling, such as metalloproteinase type I and 9, immune related cytokines (CXXL11,il 1-2-4-6-10), and growth factors (FGF, VEGF, TNF, IGF, PDGF) were all been evaluated by ELISA in order to confirm the trends obtained by means of gene expression. As reported in Figure 5b, collagene type I and vitronectin show great production for both scaffold types, as well does the anti-inflammatory cytokine IL10, confirming the excellent biological properties of the scaffold. Inflammatory-related protein such as IL1-2-4-6 and TNF were detected in all conditions. Growth factors were present at high levels with both scaffolds, although the production of VEGF, which is important during the first phase of regeneration, was higher when the support was formed by dentin. The synthesis of cxcl11, FGF, and SMAD5/8 increased, although not to the level of significance.

### 2.6. Intracellular ALP Activity

The ability of bone ECM production was investigated by means of alkaline phosphatase (ALP) activity. Bone ECM mineralization requires as a first step the formation of hydroxyapatite crystals within matrix vesicles, budding from the surface membrane of odontoblasts and osteoblasts. The subsequent propagation of hydroxyapatite into the ECM and its deposition between collagen fibrils ensures the mineralization of the ECM. The inhibitors of hydroxyapatite formation include extracellular inorganic pyrophosphates, which in normal condition are hydrolyzed by ALP, providing inorganic phosphate and promoting mineralization. As reported in Figure 6, the dentin-based scaffold (DP) induced greater presence of ALP (blue bars), confirming its ability to support faster ECM mineralization.

### 2.7. Morphology

The interactions between the DPSCs and both DP and BG were investigated in depth using SEM. As shown in Figure 7, the cells were attached to both granules. After 21 days of culture, the DPSCs interacted directly with both materials, with the cells cultured on DP exhibiting a typical odontoblastic phenotype characterized by long cell processes and very short cell bodies (Figure 7A), whereas those seeded on BG were flat with an osteoblastic-like phenotype (Figure 7B). On the DP surface, the cells showed a columnar cylindrical cell body connected directly by small, thin filaments to the body of adjacent cells.

### 2.8. Compressive Properties of Extracellular Matrix

The mechanical properties of DP and BG without cells and after 21 days of cell seeding are presented in Figure 8. In the axial compression tests on DP and BG before and after 21 days of cell culture, values tended to increase over time in the DP group, while in the BG group they remained stable. The differences in ultimate compressive strength between the DP and BG samples after 21 days of culture were not significant. Similarly, no significant difference in compressive modulus was observed between DP and BG after 21 days of cell culture.

## 3. Discussion 

All cells interact with their extracellular matrix (ECM); stem cells rely on this interaction to guide their differentiation. The nano-topography, stiffness, stress protein composition, and strain inherent in any given ECM influences the commitment of a stem cell lineage [35,36,37,38,39,40,41]. This dynamic cell–ECM interaction evolves over time, and it is from this concerted interaction that tissues are formed. Mesenchymal stem cells can give rise to cartilage, bone, muscle, and tendon; while their differentiation can be manipulated, it must be consistent with the physical properties of incredibly complex native microenvironments in order to achieve regenerative goals [36,42,43,44,45,46]. For a long time, stem cells’ fate was assumed to depend entirely on biochemical messengers, while early evidence of these cells’ mechanically-induced behavior attracted less attention [47]. The independent mechanical factors guiding stem cell lineage commitment include strain [37], elasticity [38,48,49,50], and the nano-topography [39,51] inherent in the ECM. Stiffness, compressive strength, and pore size all affect the osteogenic differentiation of mesenchymal stem cells during bone healing [38,52,53]. The physical environment experienced by a resident mesenchymal stem cell through its ECM plays a major part in the cell’s ultimate fate. ECM stiffness guides the cell’s development and dictates its response to a disease or insult. In a sense, bone is the most mechanically sensitive organ, as it is constantly being remodeled in a normal mechanical environment [36]. Clinically, a lack of mechanical stimuli or exposure to external mechanical unloading are known to reduce weight-bearing bone formation and weaken a bone’s structure [54]. Several researchers have demonstrated that multifactorial regulators are involved in the mechanical stimuli that modulate osteogenic cell activity and bone formation, including hormones or cytokines, cytoskeleton proteins, signaling pathways, and microRNAs (miRNAs).

Epigenetics is a discipline seeking to define of tissue-confined transcription factors and co-regulators that interfere with gene expression of genes involved in this process and then in the odonto- and osteogenic processes, in addition to the formation, remodeling, and repair of bone. These control processes include transcriptional control of genes through modifications of chromatin and nucleosomal architecture, DNA methylation–phosphorylation and ubiquitin-mediated degradation, and miRNA-based modulation of protein levels. MiRNAs provide novel possibilities with respect to post-transcriptional control thanks to their ability to directly inhibit mRNA translation by binding to the 3′-untranslated region (3′-UTR) of target mRNA molecules. This process often leads to the degradation of the mRNA. The generation of active miRNAs starts with their formation in the nucleus, and is followed by their maturation in the cytoplasm thanks to the Dicer enzyme. Moreover, miRNAs can be incorporated into the RNA- induced silencing complex (RISC), whereby they are able to mark and suppress the translation of several mRNAs, thus regulating almost 30% of protein-coding genes and various intracellular processes. Moreover, miRNAs play important roles in the recruitment and commitment of mesenchymal stem cells (MSCs) during the bone healing process. In addition, a subgroup of miRNAs is sensitive to various mechanical stimuli. These are called “mechanosensitive” miRNAs [55,56,57], and often are involved in osteogenic and odontogenic processes. The level of these miRNAs changes in correlation with extracellular mechanical stimuli, with a concomitant alteration in mRNA levels of ECM proteins (such as collagene, osteopontin, osteocalcin, osteonectin, and dental sialoprotein), enzymes involved in the calcification of the ECM (such as alkaline phosphatase), transcription factors for osteogenesis (such as RUNX), and intracellular regulators (such as WNT, the SMAD family, and CREB).

The cellular and molecular mechanisms underlying the impact of mechanosensitive miRNAs on bone formation have yet to be fully understood, becoming an emerging field of research. Indeed, several miRNAs are involved in these processes due to their large-scale regulatory of molecular-based networks; their elucidation may translate into new applications in tissue regeneration and modification of genetic disorders. Here, we have focused on these new molecular factors because preserving the mechanical properties of cells grown on the surface of scaffolds is crucial to the successful use of bone grafts in load-bearing applications in dentistry. Previous studies [21] have demonstrated that the use of DP has no deleterious effects on the mechanical properties of human bone and improves clinical bone regeneration thanks to its excellent osteogenic properties. In the present study, the beneficial effects of using DP as an autologous bone substitute derived from DP have been revealed at the molecular level, where it works as a potent osteogenic and odontogenic accelerator. Gene expression and morphological analyses confirmed that DPCSs were only able to acquire both osteogenic and odontoblastic phenotypes when seeded on DP surfaces. However, DPSCs produced an ECM with similar compressive strength values on both biomaterials.

Dentin sialoproteins (DSPPs) were detected on DP cultures. DSPPs are the most prevalent non-collagenous acidic proteins present in dentin play important roles during tissue mineralization, and are broadly expressed in dentin rather than in bone. DSPP shares a comparable composition with osteopontin (OPN) and bone sialoprotein (BSP). Moreover, DSPPs are involved in dental pulp stem cell commitment into odontoblast-like cells, binding their integrins and in the end promoting cell migration, attachment, commitment, and mineralization of dental mesenchymal cells. DSPPs are involved in SMAD1/5/8 phosphorylation and in their nuclear translocation via ERK1/2 and P38 signalling. The SMAD1/5/8 complex binds to SMAD binding elements (SBEs). Several immunohistochemical studies with specific anti-DSPP antibodies confirm DPSS localization in odontoblasts, predentin, and dentin, where it appears close to odontoblast processes along the dentinal tubules [58,59]. Moreover, in vivo research has indicated that it could be involved in the starting process of dentin mineralization, not in the maturation of this tissue [60].

Epigenetic data have confirmed the presence of factors that improve the cells’ odontogenic and osteogenic commitment when DPSCs are seeded onto the DP surface. As shown in Figure 9, the factors for the osteogenic pathway included WNT, miRNA 26, β-catenin, TGF, alkaline phosphatase, and p38MAPK, while for odontogenic commitment the factors identified were smad1, miRNA 200, CREB, and MSX [61]. VEGF is one of the most important growth factors, promoting neoangiogenesis and thereby accelerating new bone formation [62]. Interestingly, upregulation of VEGF was observed only when cells were seeded onto DP.

## 4. Materials and Methods

### 4.1. Isolation of Dental Pulp Stem Cells

Human dental pulp was extracted from healthy molar teeth extracted from adults aged from 18 to 66 years due to pericoronitis or for orthodontic reasons. All subjects provided their written informed consent to donating their dental pulp for scientific research. The Ethical Committee of Tel Aviv University (Department of Oral Biology, School of Dental Medicine and Department of Bio-Engineering, Faculty of Engineering, Tel Aviv University, Tel Aviv, Israel) approved the research protocol. Following our previous protocol [35], on removal the pulp was immersed for 1 h at 37 °C in a phosphate-buffered saline (PBS) solution with 100 U/mL penicillin–streptomycin, 500 mg/mL clarithromycin, 3 mg/mL type I collagenase, and 4 mg/mL dispase. After digestion, the solution was filtered through 70 mm Falcon strainers (Becton and Dickinson, Franklin Lakes, NJ, USA). For each single pulp sample, cells were cultured in vitro as a monolayer up to p1 in non-differentiative non-hematopoietic (NH) stem cell expansion medium (Miltenyi Biotec, Bergish Gladbach, Germany).

### 4.2. Characterization of DPSCs by Flow Cytometry

Adherent DPSCs at passage 3 were dissociated and resuspended in flow cytometry staining buffer (R&D Systems, Minneapolis, MN, USA) at a final cell concentration of 1 × 106 cells/mL. For characterization of surface markers, the following fluorescent monoclonal mouse anti-human antibodies were used: CD29 APC (Thermo Fisher Scientific, San Diego, CA, USA), CD73 APC (eBioscience™, Thermo Fisher Scientific), CD90 BV510 (BD Biosciences, San Jose, CA, USA), CD105 PE-Cyanine7 (eBioscience™), CD14 PE (eBioscience™), and CD45 Pacific Orange (Thermo Fisher Scientific). Cells were washed twice with 2 mL of flow cytometry staining buffer and resuspended in 500 μL of flow cytometry staining buffer. Fluorescence was evaluated by flow cytometry in Attune NxT flow cytometer (Thermo Fisher Scientific). Data were analyzed using Attune NxT software (Thermo Fisher Scientific).

### 4.3. Sample Preparation with Kometa Bio^®^

DP in granules was prepared and processed with the Kometa Bio^®^ tool (Smart Dentin Grinder-SDG, New Jersey, NY, USA) according to the manufacturer’s instructions, as follows:Remove crowns and fillings of any kind, dental plaque, discolored dentin, decay, and periodontal ligament, and rinse with a sterile physiological saline;Crush tooth fragments with the SDG (Kometa Bio^®^) to obtain and collect DP with a size between 300 and 1200 μm and with a porosity of 2–14 μm;Wash the DP for 10 min in a sterilized glass container with a 0.5 molar solution of NaOH with 20% ethanol, followed by 10 min in a sterile saline solution.

The enamel was not removed.

### 4.4. Deproteinized Bovine Bone Matrix

Hydroxyapatite (HA)-based scaffolds made of deproteinized bovine bone matrix were supplied in granules 250–1000 μm in size (Bio-Oss^®^, Geistlich Pharma Ag, Wolhusen, Switzerland). This biomaterial is hereinafter referred to as bone graft (BG).

### 4.5. Cell Culture

DPSCs were cultured in vitro as a monolayer in complete Dulbecco’s Modified Eagle Medium (cDMEM) up to step 1. At confluence, the cells were harvested using a trypsin treatment and seeded onto DP and BG samples previously anchored to the bottom of the plates with 3 mL of gelatin (Sigma) at a density of 106 cells/cm^2^. The cells were cultured for 7 days in cDMEM, changing the medium twice during the week.

### 4.6. Cell Proliferation

To assess the cell proliferation rate of DPSCs on both DP and BG, methyl thiazolyl tetrazolium (MTT) assays were performed. For each sample, the culture medium was removed and the granules with the cells were incubated at 37 °C for 3 h in 1 mL of MTT solution (0.5 mg/mL MTT in PBS), then the MTT was extracted with 0.5 mL of 10% dimethyl sulfoxide in isopropanol and the generated amount of blue formazan was determined. The optical density (O.D.) was recorded in duplicate for each sample at 570 nm in 200 μL aliquots using a multilabel plate reader (Victor 3, Perkin Elmer, Milan, Italy). Samples were examined at four different time points (after 3, 7, 14, and 21 days of culture).

### 4.7. Real-Time Polymerase Chain Reaction (RT-PCR)

Following the manufacturer’s instructions, RNA was extracted from cells seeded onto DP and BG for 21 days using a Total RNA Purification Plus Kit (Norgen Biotek Corporation, Thorold, ON, Canada). The total RNA obtained from the specimens was reverse-transcribed with a SensiFAST™ cDNA Synthesis kit (Bioline GmbH, Konstanz, Germany) in a LifePro Thermal Cycler (BioerTechnology, Hangzhou, China), with 10 min of annealing at 25 °C, 45 min of reverse transcription at 42 °C, and 5 min of inactivation at 85 °C. Human primers had been selected in advance for each target gene using Primer 3 software (Thermo-Fisher Scientific, Berlin, Germany). RT-PCR was performed with the selected primers at a concentration of 400 nM using a SensiFAST™ SYBR No-ROX kit (Bioline GmbH, Luckenwalde, Germany) on a Rotor-Gene 3000 cycler (Corbett Research, Sydney, Australia). The list of utilized primers and related sequences is presented in Table 1. For miRNAs, we followed the instruction of a Human miRNA Microarray Kit, V3 (Agilent Technologies), with 866 sequences for human miRNA. For genes related to extracellular matrix and adhesion proteins, an RT² Profiler PCR Array kit was utilized (Qiagen, Hilden, Germany). The array was performed three times on three different preparations.

The thermal cycling conditions were as follows: 2 min of denaturation at 95 °C, followed by 40 cycles of denaturation for 5 s at 95 °C, annealing for 10 s at 60 °C, and elongation for 20 s at 72 °C. Data analysis was carried out following the 2^−ΔΔCt^ method.

The threshold cycle (Ct) values (i.e., the fractional cycle number at which the amount of amplified target reaches a given threshold) of the target genes were normalized to a housekeeping gene (transferrin receptor 1). Gene expression was compared between the control group (DPSCs on plastic monolayer) and the test groups (DPSCs cultured on DP and BG). The results are reported as the fold regulation of target genes in the test groups compared with the DPSCs on the monolayer. The fold regulation indicates increased (values > 2) or decreased (values < −2) gene expression, while values between −2 and 2 indicate differentially expressed genes.

### 4.8. Intracellular ALP Activity Assay

The intracellular alkaline phosphatase (ALP) activity was detected with a colorimetric Alkaline Phosphatase Assay Kit (Abcam, Cambridge, UK). DPSCs seeded onto DP and BG for 7, 14, and 21 days were washed with PBS, homogenized with ALP Assay Buffer, then centrifuged at 13,000 rpm for 3 min to remove insoluble material. A standard curve was drawn using the corrected absorbance values of the standards, and the pNP concentration was identified for each sample. ALP activity was calculated as follows: ALP activity (U/mL) = (*A*/*V*)/T, where *A* represents the amount of pNP generated by samples (in μmol), *V* is the volume of sample added to the assay well (in mL), and T is the reaction time (in min).

### 4.9. Scanning Electron Microscopy (SEM)

For SEM imaging, DPSCs grown on DP and BG for 21 days were fixed in 2.5% glutaraldehyde and 0.1 M cacodylate buffer for 1 h, then gradually dehydrated in ethanol. SEM analyses were conducted with a JSM JEOL 6490 SEM microscope (JEOL, Tokyo, Japan) at the Centro di Analisi e Servizi per la Certificazione (CEASC, University of Padova, Padova, Italy).

### 4.10. Compressive Properties of Extracellular Matrix

The compressive strength and compressive modulus of each grafting material were measured using a Hounsfield H10K/M527 testing machine (Hounsfield, NY, USA) before and after 21 days of DPSC culture inside the granules. The grafting materials were reasonably homogeneous, and thus the orientation of the compressed surfaces was not recorded. The granules were gently removed from the wells and placed inside of a flat ring. Rubber pads were placed on the top and bottom surfaces of each sample to ensure an evenly-distributed load.

To prepare the samples for mechanical compressive strength testing, the granules were placed inside the flat ring (diameter 8 mm × height 2 mm), then the ring was carefully removed. The crosshead speed was set at 1mm/min and the load was applied until the sample was crushed completely. Between seven and nine identical specimens from each sample group were used for the compression tests.

### 4.11. ELISA Assay

Enzyme-linked immunosorbent assays (ELISA) (SIGMA, Aldrich, Berlin, Germany) were performed to detect the presence of the following proteins: collagen I, vitronectin, matrix metalloproteinases (MMP1, MMP9), CXCL 11, several interleukins (IL1, IL2, IL4, IL6, IL10), fibroblast growth factor (FGF), vascular endothelial growth factor (VEGF), tumor necrosis factor (TNF), insulin-like growth factor (IGF), platelet-derived growth factor (PDGF), DSPP, MSX, SMAD1, CREB, BMP2, and SMAD 1/5/8.

### 4.12. Statistical Analysis

One-way analysis of variance (ANOVA) was used for data analyses. Data were expressed as mean ± standard deviation (SD). A *p*-value < 0.05 was considered to be statistically significant.

## 5. Conclusions

Because bone is a living tissue, the form and function of which depend on the composition and organization of its matrix and on its structural features, any interference with the processes involved in bone repair can prevent its regeneration. Bone tissue is incapable of interstitial growth, which is why bone mass is governed by complex phenomena. Changes in the mineralized matrix of bone can only be made by osteoblasts and osteoclasts, the behavior of which is governed by osteocytes, which are embedded master regulators of the bone formation and resorption processes [62]. When a population of bone cells received a mechanical signal, this is translated into a first biological signal. Then, a series of secondary biochemical signaling events have to occur in order to propagate the signal through the cells and to other sensor/effector cells. Research on the signaling pathways involved in mechanical signal propagation through bone cells has identified a multitude of changes that take place in mechanically-stimulated osteocytes and/or osteoblasts. These changes can affect gene expression, proteins, and lipids (e.g., phosphorylation events), can prompt protein degradation, intracellular translocation events, and the release of secreted factors, and can even alter a cell’s shape and size, among other effects. The challenge posed by these research findings is to identify which of these events are needed for mechanotransduction to occur and which events have few functional consequences for this process [63,64,65]. DP induces a mechanoinduction that takes effect on the stem cells’ commitment, stimulating both the osteogenic pathway (by means of WNT signaling, with the expression of miRNA26-27-32-586-885 and SMAD4) and odontogenic phenotyping (thanks to miRNA 200, miRNA 218, CREB and SMAD1). Moreover, protein secretion analyses confirm the presence of a higher amount of growth factors related to angiogenesis, such as VEGF, on DP compared to BG. In any case, the expression of bmp2 and smad5/8 supports the evidence that the 3D scaffolds support osteogenic commitment.

This protein, the presence of which could derive from either DPSC activity or the granule content, can provide a stronger support during bone regeneration, as the formation of novel vessels is crucial for driving proper tissue remodeling. Furthermore, the high levels of ALP detected only in the presence of DP could be explained as deriving from dentin matrix. In this view, dentin granules may be considered as a biological reservoir of the necessary mineral ions, growth factors, and ECM molecules, providing a favorable bioinductive microenvironment and thereby affecting extracellular secretoma production [66,67,68]. Cells secrete several types of extracellular vesicles that can contain a complex cargo of proteins, lipids, and nucleic acids. Their content is heterogenous and depends on the physiological conditions of the cell. Moreover, environmental changes affect the content of exosomes from cells. Thus, extracellular vesicles represent a key concept in mediating changes in cellular behaviour by affecting cells in a paracrine or an endocrine manner, which makes them useful therapeutically. This is important mainly during tissue regeneration, where it represents a major advantage as a cell-free therapy [68,69,70,71,72]. Moreover, unlike other nanoparticles, exosomes show rapid clearance and the absence of unwanted accumulation in the liver, explaining their favourable toxicity profile; moreover, they do not express HLA (human leukocyte antigen) class II, making them hypoimmunogenic. Due to their great biocompatibility, exosomes have attracted wide attention in the medical community. Along with tissue-engineering techniques, exosome-based strategies are promising for the development of improved clinical therapies. Exosomes could be used as biomimetic tools to induce migration, proliferation, and odontoblast-specific differentiation of stem cells in regenerative therapy. Based on this evidence, it is possible to speculate that dentin-based particles may have a positive effect on the differentiation of naïve cells, inducing them to produce exosomes that directly trigger odontoblastogenetic regeneration [73,74,75,76,77,78,79,80,81,82,83,84,85].

In conclusion, the ECM of DPSCs cultured on DP gives rise to a very precise cell–matrix connection and creates a mechanical transduction pathway, mostly via epigenetic changes, that is focused particularly on specific mechanosensitive miRNAs.

## Figures and Tables

**Figure 1 ijms-23-09283-f001:**
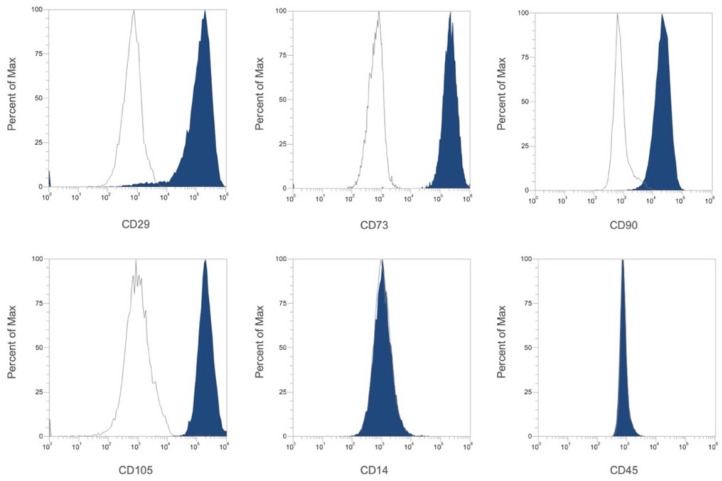
Characterization of cell surface markers in DPSCs at passage 3 by flow cytometry. The stem cells isolated from dental pulp are positive to CD29, CD73, CD90, and CD105 MSC-specific markers and negative to CD14 and CD45 hematopoietic markers. White negative, Blue positive.

**Figure 2 ijms-23-09283-f002:**
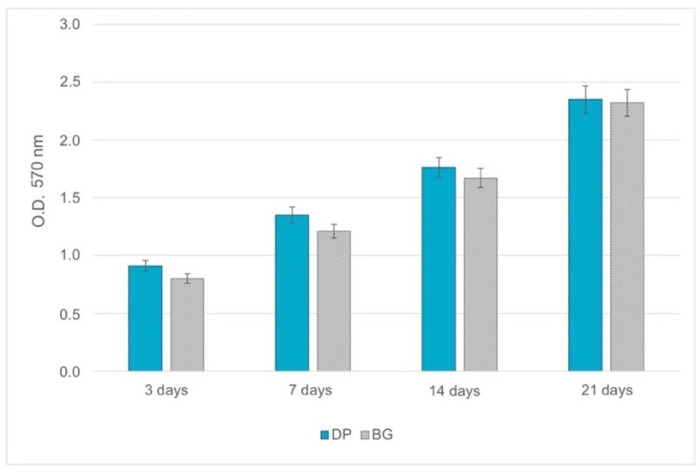
MTT assay on DP (petrol blue bars) and BG (grey bars) granules. Tests were run from 3 to 21 days of culturing. The graph shows the mean ± SD of three different experiments.

**Figure 3 ijms-23-09283-f003:**
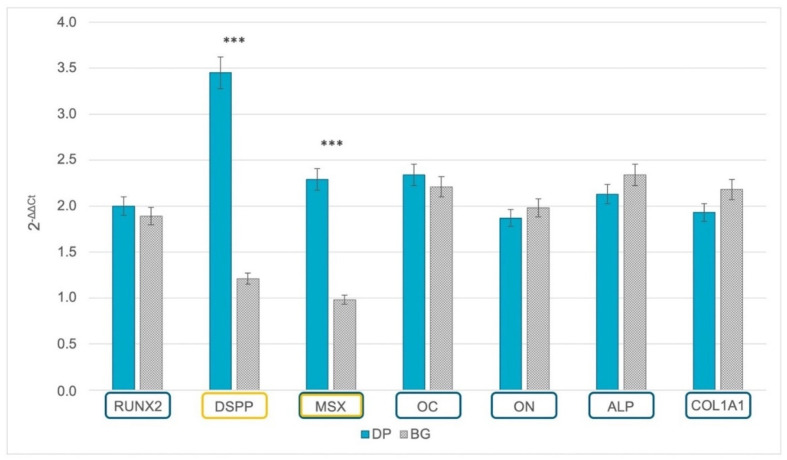
Real-time PCR of odontogenic and osteogenic markers highlighted in orange and blue, respectively. Gene expression levels of the selected markers are compared between the samples derived from DP (petrol blue bars) and BG granules (grey bars) after 21 days of culture. The graph shows the mean ± SD of three different experiments *** *p* < 0.001. Fold obtained with compared to the related value of cells cultured onto plastic supports. Time of culture was 21 days.

**Figure 4 ijms-23-09283-f004:**
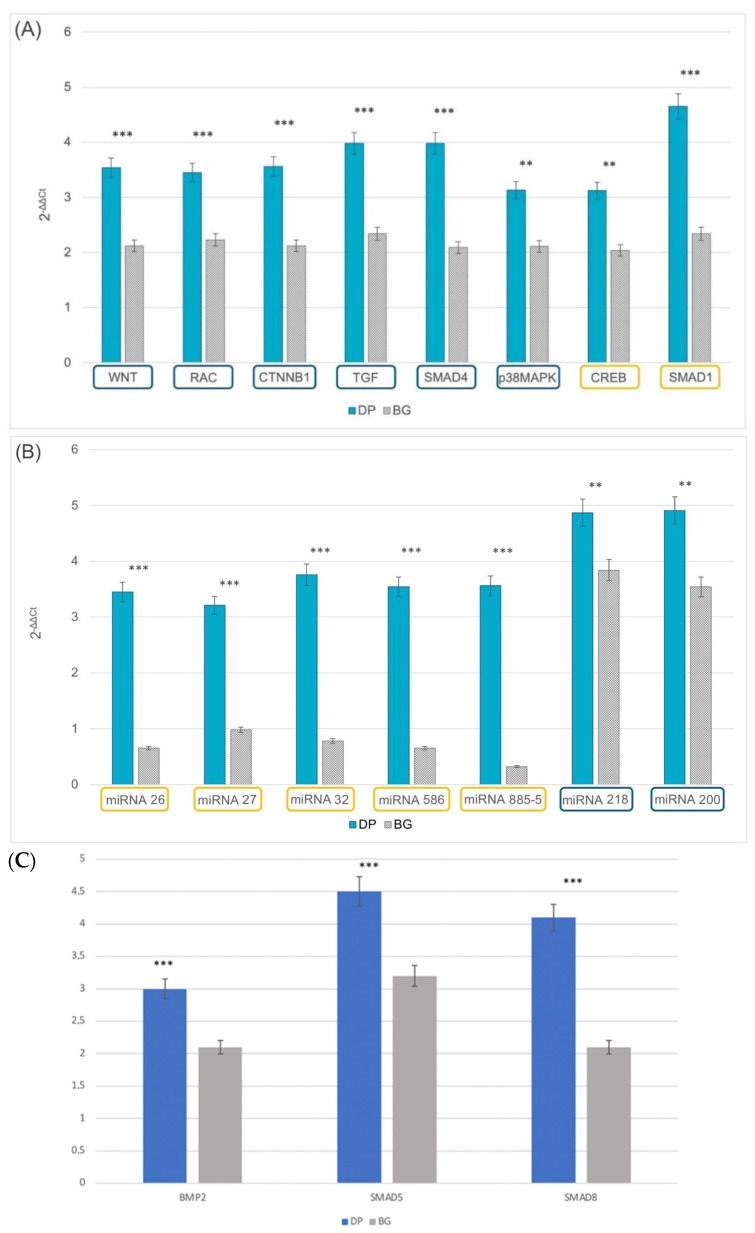
Real-time PCR of genes for proteins (**A**), miRNAs (**B**), and mRNA (**C**) involved in odontogenic or osteogenic commitment of DPSCs cultured up to 21 days on DP (petrol blue bars) or BG (grey bars). Genes involved in odontogenic commitment are highlighted in orange, while those associated with osteogenic commitment are in blue. The graphs show the mean ± SD of three different experiments. Fold obtained compared to the related value of cells cultured onto plastic supports. Time of culture 21 days. ** *p* < 0.01, *** *p* < 0.001.

**Figure 5 ijms-23-09283-f005:**
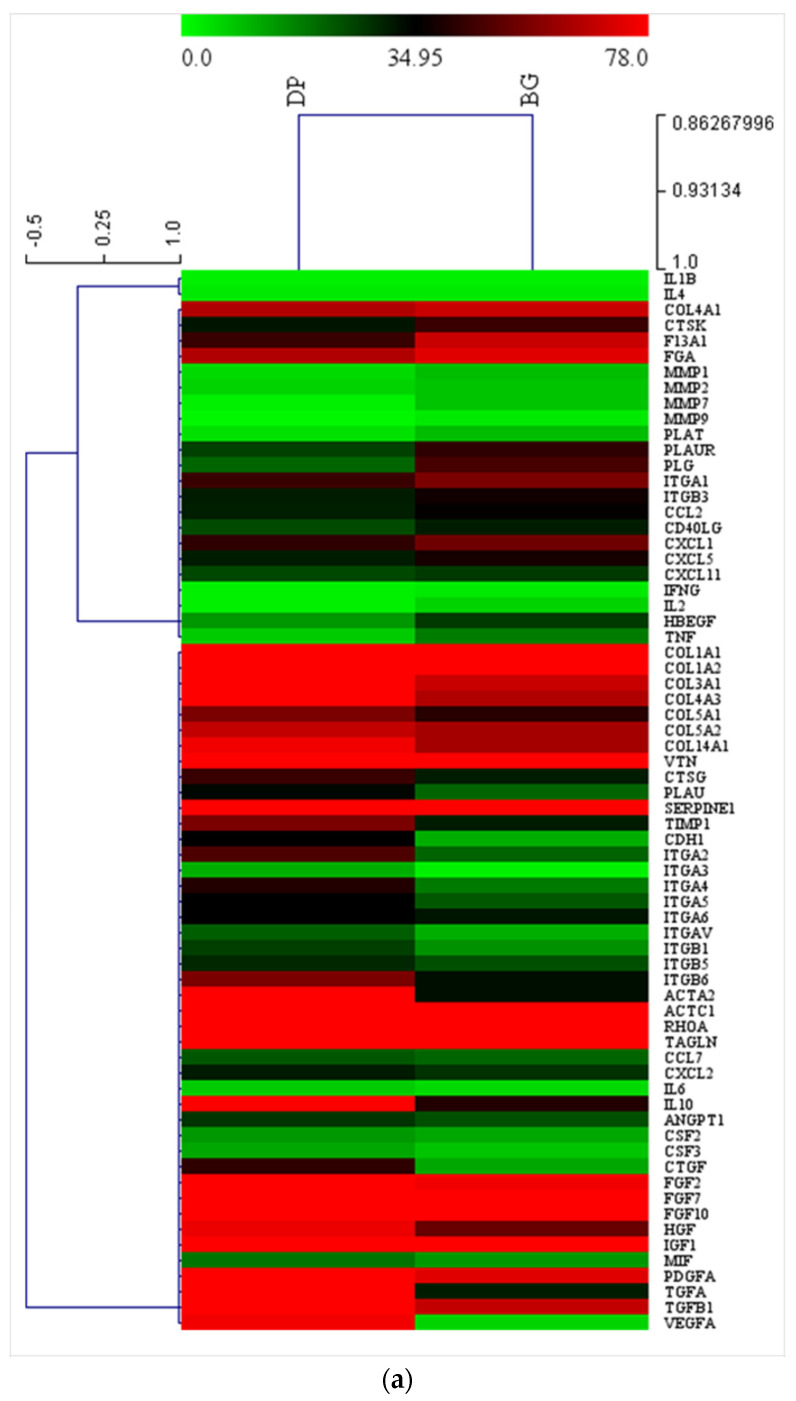
(**a**) Graphical representation of real-time PCR of mRNA related to genes for ECM and adhesion proteins of DPSCs cultured up to 21 days on DP (petrol blue bars) or BG (grey bars); (**b**) ELISA assay of proteins expressed by DPSCs cultured up to 21 days on DP (left) or BG (right). *** *p* < 0.001.

**Figure 6 ijms-23-09283-f006:**
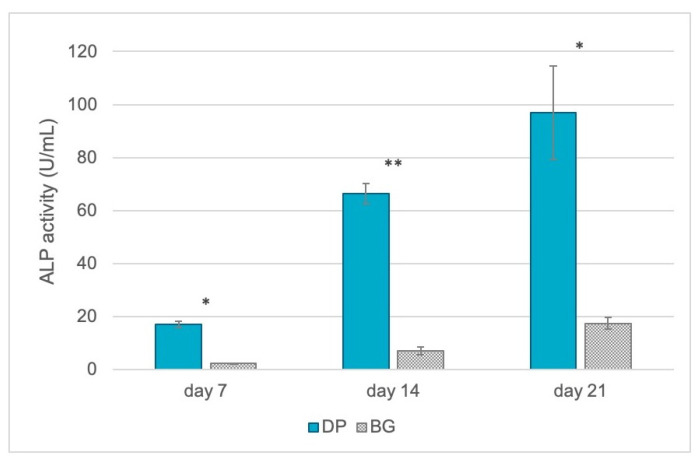
Quantification of intracellular ALP activity (expressed as U/mL) in DPSCs seeded for 7, 14, and 21 days onto DP (petrol blue bars) or BG (grey bars). The graph shows the mean ± SD of three different experiments. * *p* < 0.05, ** *p* < 0.01.

**Figure 7 ijms-23-09283-f007:**
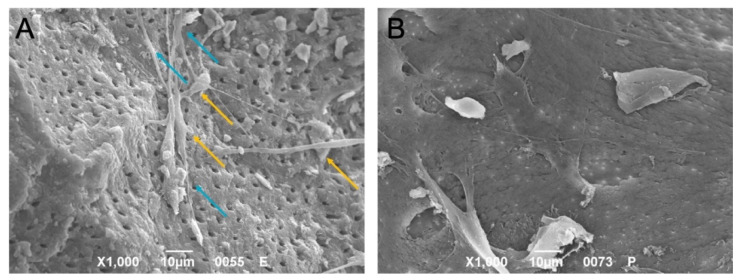
SEM analyses of DPSCs seeded on DP or BG granules ((**A**) and (**B**), respectively). When cells were seeded onto DP, they acquired an odontoblastic phenotype characterized by a short body (orange arrows) and an elongated cell process (petrol blue arrow).

**Figure 8 ijms-23-09283-f008:**
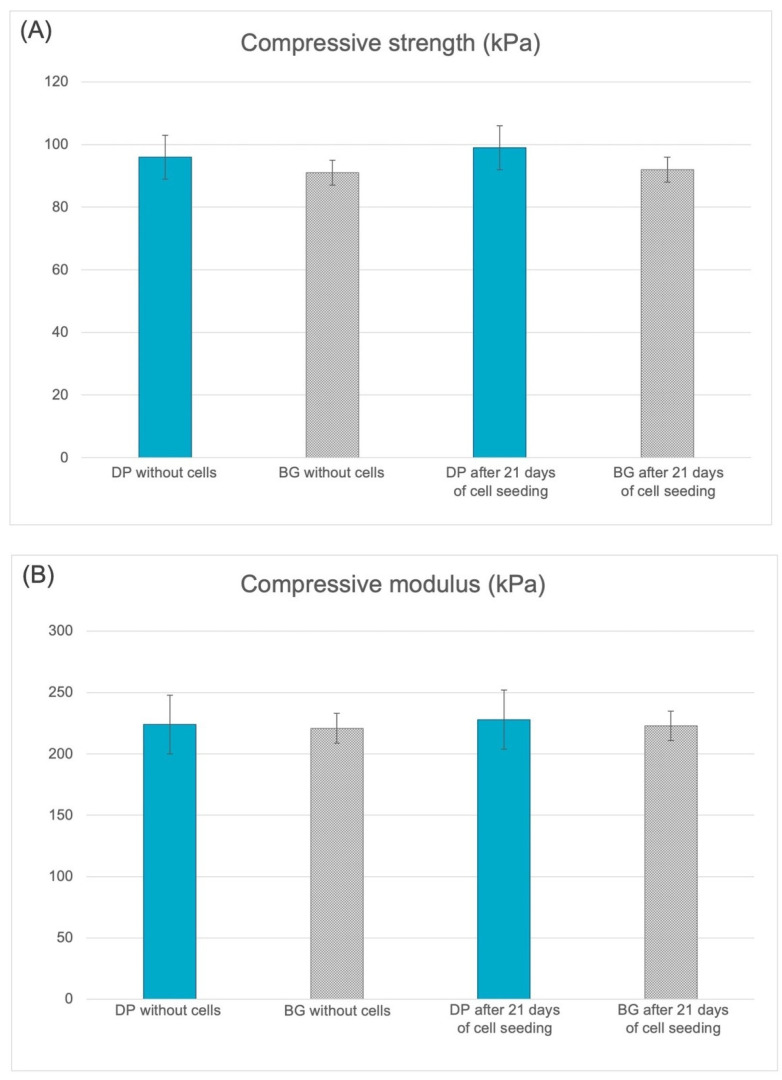
Compressive strength (**A**) and compressive modulus (**B**) of DP and BG without cells and after 21 days of cell seeding.

**Figure 9 ijms-23-09283-f009:**
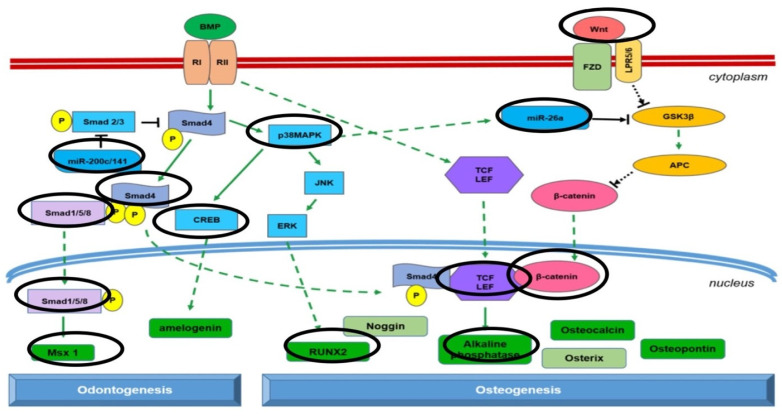
Pathways expressed in DPSCs cultured on dentin surfaces. DPSCs cultured on DP gave rise to the production of a harder ECM than that obtained from xenogenic bone grafts. This greater external mechanoinduction acts on the commitment of stem cells, stimulating both the osteogenic pathway (by means of WNT signaling with the expression of miRNA 26a), and odontogenic phenotyping (thanks to miRNA 200 and SMAD1). Modified from Zhang et al. [61].

**Table 1 ijms-23-09283-t001:** Human primer sequences.

Gene Symbol	Forward Primer (5′→3′)	Reverse Primer (5′→3′)	Product Length (bp)
ALP	TCAGAGGGAAGGAGATAGAGAGTC	AGCCAGAAACCATATGTCAAGAGA	171
COL1A1	GAGCCAGCAGATCGAGA	ACCAGTCTCCATGTTGCAGA	178
DSPP	AGATCCGCACGCAGTAT GAG	AGGTCGCAGGTCAAGGA	178
MSX	CAGGAGATCACAGAGTATGCCAA	AGATGCGGTGGCTAAAGGTC	179
OC	TGCATGTGTCTTAGTCTTAGTCACCGCTA	ACTTAGTGCTTACAGGAACCA	167
ON	TGCATGTGTCTTAGTCTTAGTCACC	GCTAACTTAGTGCTTACAGGAACCA	186
RUNX2	TCTTAGGCAGCTCTTTGGGA	TCCCTTGTCATGAAGCCTTGG	182
CTNNB1	TAATAAACAGCTCTAAGCCCA	ATCCTCTACATTTAGCCTAGA	176
CREB	CGTTAGGCGGCTCAATGGGA	TAACTCTTCCGGAAGCCTAAG	178
p38MAPK	GTTCCGGCTTGCTCTTTGCTA	TCCAATGACATCCGAGCCTAGA	181
RAC	AACTCATTCCTTTGCCTT	TCAATGATGTTCCCTCCAG	184
SMAD1	GGATAGAGGCTTTGGGACCT	TCCTCTTGAAGGGTCCTTGCA	178
SMAD4	TGCATTACGATCAAGGCTG	TATTGGATTGGAAGCTGCCCTTG	167
TGF	CCGGGCAGAATTTGA	TGGTCAATGTGAAGAA GG	185
WNT	GCTCTTCAAAGGTATTTG	TGGTCCAACCCGTCATC CT	185

Gene symbols and related encoded proteins are listed here. ALP: alkaline phosphatase; COL1A1: collagen type I; CREB: cyclic AMP response element binding protein; CTNNB1: β-catenin; DSPP: dentin sialoprotein precursor; MSX: muscle segment homeobox gene family; OC: osteocalcin; ON: osteonectin; p38MAPK: p38 mitogen-activated protein kinases; RAC: Rac; RUNX2: runt-related transcription factor 2; SMAD1/4: smad1/4; TGF: transforming growth factor; WNT: Wnt.

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
