# Peer review of "Dentin Particulate for Bone Regeneration: An In Vitro Study"

_ijms, 2022, doi:10.3390/ijms23169283_

Round 1
Reviewer 1 Report
- Add figure 4c to 4A. provide statistics
- Add results of bmp2, smad5/8 to abstract and to text in section 2.4
- In line 193, authors say “inflammatory related protein such as IL1-2-4-6 and TNF have been detected.” But in the revision letter they say it was close to zero. Please correct the text or show exact values
- What is authors interpretation of increased bmp2, smad5/8? Add to results and discussion.
- Fig 5b, cxcl11, FGF, SMAD5/8, were not significant?
- Fig5b, what is vertical axis? Add to graph
- Fig 3 and 4a-c, how the fold change is calculated? Compared to what? Day 0? Day 21? Add to chart legend
Author Response
Dear Referee thanks for your suggestion that strongly improves the quality of the paper:
Add figure 4c to 4A. provide statistics
thanks done
- Add results of bmp2, smad5/8 to abstract and to text in section 2.4
- thanks done
- In line 193, authors say “inflammatory related protein such as IL1-2-4-6 and TNF have been detected.” But in the revision letter they say it was close to zero. Please correct the text or show exact values
- thanks done
- What is authors interpretation of increased bmp2, smad5/8? Add to results and discussion.
- thanks done
- Fig 5b, cxcl11, FGF, SMAD5/8, were not significant?
- we added a comments
- Fig5b, what is vertical axis? Add to graph
- thanks done
- Fig 3 and 4a-c, how the fold change is calculated? Compared to what? Day 0? Day 21? Add to chart legend
- thanks improved
Reviewer 2 Report
Dear authors,
the changes made to this article have made it more correct and understandable in some points.
I therefore consider it deserving of being published in the International Journal of Molecular Sciences.
Best regards
Author Response
thanks so much
Round 2
Reviewer 1 Report
-
This manuscript is a resubmission of an earlier submission. The following is a list of the peer review reports and author responses from that submission.
Round 1
Reviewer 1 Report
Dear Authors,this article is well written and of good scientific interest, and deserves to be published in the International Journal of Molecular Sciences.
I do not think it is necessary to make any changes to the text, however I suggest you to check the spelling carefully for the presence of numerous errors.
Best regards
Author Response
Thanks to the referee we revised the article according to his suggestion
Reviewer 2 Report
- In line 72, provide examples of those studies
- Move line 82-84 to the beginning of line 75.
- In section 4.1, the culture of cells may be added
- In section 4.3, was enamel removed?
- Section 4.5, explain and provide as much details as possible about cell culture. How DPs were placed in culture plate? They were not attached to plate, so they could move and disrupt cells during media change. Provide figures of DPs and BG
- Section 4.5, why the cells were detached and seeded again after 7 days?
- Section 4.7, was DP separated before putting in RNA isolation buffer? Is there a chance that DPs may contribute and add to DSPP detection?
- In 4.8, was equal amount of protein used for all samples?
- Provide figures for 4.10
- In figure 3, mention the time point from which there results were derived. i.e. 21 days?
- Section 2.4, how the markers for epigenetics for selected? Provide reference
- Western blots for DSPP and MSX in figure 3 and for SMAD 1 and CREB in figure 4 should be provided.
- In figure 4A, BMP2 and SMAD1/5/8 should be provided
- In figure 5, there are no blue and grey bars as mentioned in the legend. Correct it.
- In figure 5, was this a result of microarray analysis? Provide details in methods section
- Is it possible to provide confocal images of the 2 groups to look it their cytoskeleton and/or ECM arrangement?
- In all figures, mention the time point so the reader does not have to go and check the text
- In 2.8, table 2 is missing
- I suggest looking at secretome of cells culture on the 2 groups and looking at miRNA cargo of EVs or doing some proteomics of what is observed in gene expression levels.
- In section 239, when talking about epigenetics and miRNA, mention the possible role of EVs and their cargo in this process. Refer to and cite “the importance of cellular and exosomal miRNAs in mesenchymal stem cell osteoblastic differentiation." Scientific reports 11.1 (2021): 1-14.”
- Discussion has a lot of sentences that talk about the surrounding topics of this study, however, they are not directly related to what is observed in the current study. Either summarize it, or relate it to your findings in more details and discuss your findings more
- Line 364, Conclusion is not supported by your data. No pathway analysis or RNA seq is perfumed to support these claims. Please modify
Author Response
Dear Referee
Thanks for your comments that strongly improve the quality of the article.
Ref N°
- in line 72, provide examples of those studies
thanks done
- Move line 82-84 to the beginning of line 75.
thanks done
- In section 4.1, the culture of cells may be added
thanks done
- In section 4.3, was enamel removed?
no It is not
- Section 4.5, explain and provide as much details as possible about cell culture. How DPs were placed in culture plate?
We improved the section. We used a tin layer of gelatin and
- They were not attached to plate, so they could move and disrupt cells during media change. Provide figures of DPs and BG
ok
Section 4.5, why the cells were detached and seeded again after 7 days?
it was an error
- Section 4.7, was DP separated before putting in RNA isolation buffer?
No they do not
- Is there a chance that DPs may contribute and add to DSPP detection?
No, because our control was DP without cells and we isolated any RNA useful for PCR
In 4.8, was equal amount of protein used for all samples?
Yes it is
In figure 3, mention the time point from which there results were derived. i.e. 21 days?
- Yes
- Section 2.4, how the markers for epigenetics for selected? Provide reference
done
- In figure 5, there are no blue and grey bars as mentioned in the legend. Correct it.
- Done, Thanks
- In figure 5, was this a result of microarray analysis? Provide details in methods section
No it is not. It is a heat map of a PCR analyses from a kit containing several mRNA as reported on material and methods section, that we improved.
- Is it possible to provide confocal images of the 2 groups to look it their cytoskeleton and/or ECM arrangement?
no the confocal is not able to perform an image of the cytoskeleton of the cells seeded onto the granules
- In all figures, mention the time point so the reader does not have to go and check the text
- done
- In 2.8, table 2 is missing
- added
- I suggest looking at secretome of cells culture on the 2 groups and looking at miRNA cargo of EVs or doing some proteomics of what is observed in gene expression levels.
- thanks we ll take into account this possibility for the next publication
- In section 239, when talking about epigenetics and miRNA, mention the possible role of EVs and their cargo in this process. Refer to and cite “the importance of cellular and exosomal miRNAs in mesenchymal stem cell osteoblastic differentiation." Scientific reports 11.1 (2021): 1-14.”
- thanks done
- Discussion has a lot of sentences that talk about the surrounding topics of this study, however, they are not directly related to what is observed in the current study. Either summarize it, or relate it to your findings in more details and discuss your findings more
- done
- Line 364, Conclusion is not supported by your data. No pathway analysis or RNA seq is perfumed to support these claims. Please modify
- done

Round 2
Reviewer 2 Report
- In the response letter, the authors must mention where in the manuscript they have added the answers. With line numbers
- Add answers to all of my questions to the text as they are needed for clarifications. Ex: “In section 4.3, was enamel removed?” add to the text that enamel was not removed
- Why the sections 4.11 and 4.12 are added? There was no use of EVs in the manuscript. Please clarify
- The possible roles of EVs and their miRNA cargo was asked to be added to the manuscript as a possible explanation for the results. please add
The authors should provide the bellow results from my previous comments
- Western blots for DSPP and MSX in figure 3 and for SMAD 1 and CREB in figure 4 should be provided.
- In figure 4A, BMP2 and SMAD1/5/8 should be provided
Author Response
Add answers to all of my questions to the text as they are needed for clarifications. Ex: “In section 4.3, was enamel removed?” add to the text that enamel was not removed
added
- Why the sections 4.11 and 4.12 are added? There was no use of EVs in the manuscript. Please clarify
- it was an error
- The possible roles of EVs and their miRNA cargo was asked to be added to the manuscript as a possible explanation for the results. please add
added
The authors should provide the bellow results from my previous comments
- Western blots for DSPP and MSX in figure 3 and for SMAD 1 and CREB in figure 4 should be provided.
we performed ELISA, we added it
- In figure 4A, BMP2 and SMAD1/5/8 should be provided
- added

Round 3
Reviewer 2 Report
- Line 179, referring to fig 6 is not correct. It should be fig 5b.
- Fig 5b, there are 2 series of Smad1. Which one is correct? Why BMP2 is not shown?
- Why did authors provide ELISA not Westernblots?
- Why gene expression levels of BMP2 and SMAD1/5/8 is not provided?
- Fig 5b, provide statistics to show the significance
- Fig 5b, I suggest showing the rows with small numbers in a separate graph so that their values are observed.
- Put 4.12 before 4.11
d - provide references to lines 346-356
- Add references to line 357-375. Cite bellow and other studies:
Extracellular Vesicles from TNFα Preconditioned MSCs: Effects on Immunomodulation and Bone Regeneration. Frontiers in Immunology, 2019
3D Encapsulation and tethering of functionally engineered extracellular vesicles to hydrogels. Acta biomaterialia 126, 199-210
The importance of cellular and exosomal miRNAs in mesenchymal stem cell osteoblastic differentiation. Scientific reports 11 (1), 1-14
Extracellular MicroRNA Expression In Gingival Crevicular Fluid During Tooth Movement. CONTROVERSIAL TOPICS IN ORTHODONTICS: CAN WE REACH CONSENSUS? 1001, 17
Macrophage Control of Incipient Bone Formation in Diabetic Mice. Frontiers in Cell and Developmental Biology 8, 596622
Bone regeneration is mediated by macrophage extracellular vesicles. Bone 141, 115627
Functionally engineered extracellular vesicles improve bone regeneration. Acta biomaterialia 109, 182-194
Advances in orthodontic tooth movement: gene therapy and molecular biology aspect. Current Approaches in Orthodontics